# Using Matchboxes to Teach the Basics of Machine Learning: an Analysis of (Possible) Misconceptions

**Erik Marx**[1]  **Thiemo Leonhardt**[1]  **David Baberowski**[1]  **Nadine Bergner**[1]

## Abstract

The idea of chess-playing matchboxes, conceived by Martin Gardner as early as 1962, is becoming more and more relevant in learning materials in the area of AI and Machine Learning. Thus, it can be found in a large number of workshops and papers as an innovative teaching method to convey the basic ideas of reinforcement learning. In this paper the concept and its variations will be presented and the advantages of this analog approach will be shown. At the same time, however, the limitations of the approach are analyzed and the question of alternatives is raised.

## 1. Introduction

As Machine Learning (ML) has an increasing influence on many people's everyday life, (Nayak & Dutta, 2017) there is a need for concepts to include ML in workshops and other learning scenarios. Building these scenarios, so that beginners can gain insight into the underlying concepts, while avoiding oversimplifications and misconceptions can be very challenging, especially when talking about recent technological developments. Unplugged materials like the Hexapawn game show a creative and motivating approach to the topic. In this paper, we will take a look at these materials and discuss what misconceptions can occur, using this approach.

## 2. Misconceptions

The study of misconceptions in computer science education and in education in general is a well-studied and still current research topic especially in computer science didactics (Sorva, 2012) (Ohrndorf, 2016) (Qian & Lehman, 2017). Misconceptions can be understood as entrenched systematic errors (Prediger & Wittmann, 2009). However, *Qian*

*& Lehmann* also discuss other hard-to-define concepts such as difficulties, mistakes, and bugs, stating that there is no single definition (Qian & Lehman, 2017). As a definition for misconceptions in CS programming education (Sorva, 2012) states the following: "understandings that are deficient or inadequate for many practical programming contexts". In reference to (Ohrndorf, 2016) we define misconceptions as cognitive representations of knowledge that contradict or deviate from the scientifically correct concepts.

(Heuer et al., 2021) examined machine learning tutorials for misconceptions and misleading explanations, identifying four main misconceptions: (H1) ML as adapting in response to new data and experiences to improve efficacy over time; (H2) ML as automating and improving the learning process of computers based on their experiences without any human assistance; (H3) ML can discover hidden patterns that are invisible to humans; (H4) ML can be applied without special expertise.

## 3. The Game

The original game idea of (Gardner, 1962) is a chess variant called "Hexapawn". On a 3x3 chess board, three pawns of different colors face each other (*Figure 1*). The objective is either to move a pawn to the opponent's baseline or to capture all of the opponent's pawns. One can also win by achieving a position in which the enemy cannot move (Stalemate). The pawns move one step forward as in chess and capture diagonally forward.

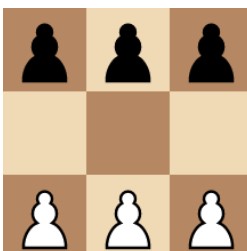

*Figure 1.* Starting position of Hexapawn

With these simple game rules it is possible to construct what

---

[1]Didactics of Computer Science, TU Dresden, Dresden, Saxony, Germany. Correspondence to: Erik Marx <erik.marx@tu-dresden.de>.

*Proceedings of the $2^{nd}$ Teaching in Machine Learning Workshop*, PMLR, 2021.

*Gardner* described as a "learning machine" or "Hexapawn Educable Robot" (HER)[1]. The human player controls the white pieces, the AI the black ones. For each position that black might face, there is a corresponding matchbox. It contains different colored beads, each of which is assigned to one particular move, that is possible in this position (*Figure 2*). White begins to play and makes the first move. When it is the AI's turn, the move is determined by picking a beat (and, subsequently, a move) randomly from the matchbox, that corresponds to the current state of the game. After that the human player makes the next move, changing the state of the game from which the next move of HER is derived.

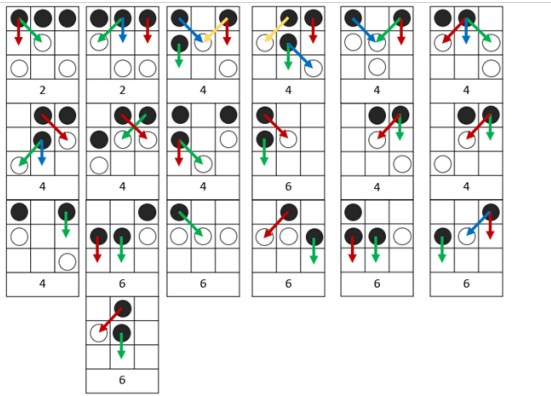

*Figure 2.* Possible positions with corresponding moves. Note that the number of boxes can be reduced due to symmetry

The reason why Gardner's idea is often used when teaching machine learning is that HER can learn. Every time when HER loses a game, the last drawn bead is removed from the box to make the corresponding losing move unable to happen again, which then makes it increasingly unlikely that HER loses in the following games. After 10-15 games against a skilled player HER is nearly unbeatable when faced with optimal play (Gardner, 1962).

## 4. Reception and Adaptions

Gardner's idea can be construed as CS Unplugged activity, even though it did not appear on the original website [2]. These focus on the problem-solving nature of computer science and are popular around the world. Unplugged material is characterized by the fact that computers are deliberately avoided and instead analog work materials and a playful approach are used to introduce the basic ideas of CS.(Nishida et al., 2009)

Besides other versions (The Royal Institution, 2008)

(Demšar & Demšar, 2015), the "Sweet Learning Computer" created by *McOwan & Curzon* as part of the *cs4fn* project is the most known version of Hexapawn created as teaching material. The authors focus on the playful aspect of the idea, for example by replacing beads with sweets, which may then be eaten in the learning step. Within the material, the workings of the Matchbox computer are described, but not the basic ideas of ML (McOwan & Curzon). Starting from this basic concept, several works can be found that convey individual aspects of ML or AI. (The Royal Institution, 2008) for example, used Gardner's example in its Christmas lecture, explaining how the machine learns. Motivated by the example, it then gives an overview of how a real chess computer works. A deeper insight into ML is provided in the material of *Lindner & Seegerer* and *Opel et. al.*, where Hexapawn is used directly to convey two principles form the field of AI (Lindner & Seegerer, 2020). The original concept is used to teach reinforced learning. Here the students should learn that computers learn by "reward" and "punishment", adapt their strategy accordingly and try to maximize their profit. For this purpose, the rules are slightly adapted by adding beads to winning moves. Second, Hexapawn is also used to explain expert systems and then compare the two methods.

(Opel et al., 2019) have created a larger set of materials[3] for an entire ML and AI module in which HER took the central role to explain and to reflect how ML works. The material was developed for students from the age of 12 and has been adapted accordingly. First and foremost, a role system and a game flow chart were created to make the learning process more understandable. In addition, questions were designed to reflect the insights gained from the game afterwards. Additionally background knowledge on ML, artificial neural networks and "deep learning" was provided to help teachers answer possible queries from students.

## 5. Limitations and Possible Misconceptions

In order to work out possible misconceptions, it is required to analyze how HER is structured. A fully trained HER formally consists of a handful of formal IF-THEN rules (if the game is in state a, then use move b). HER is therefore a symbolic AI, or more precisely a rule-based expert system (Stuart, 1992, p. 3). While these rule-based systems are very popular, they generally do not have the ability to expand their knowledge base through ML (Ogidan et al., 2018).

Thus, the attempt to represent aspects of ML by Hexapawn faces the fundamental problem that Gardner's model does not learn by common means of ML. The authors argue that

---

[1]In the following Hexapawn and HER are used synonymously

[2]https://classic.csunplugged.org/activities/

[3]https://www.wissenschaftsjahr.de/2019/fileadmin/user_upload/Wissenschaftsjahr_2019/Jugendaktion/WJ19_LA_Material_Buch_CPS_barrRZ.pdf (german)

this inconsistency can lead to the following misconceptions:

M1) ML only produces results when the complete problem/data space is exhausted.

M2) The way ML works is that undesirable behavior is trained out through negative examples.

M3) The way ML works in rule-based expert systems is that wrong rules are sorted out through negative examples.

M4) A decision tree is trained by training its nodes individually.

These misconceptions are divided into two areas. The first involves misconceptions about the field of ML and its understanding. The second one contains misconceptions about individual models used in ML.

## 5.1. Misconceptions of ML

The following misconceptions arise from the structure and learning process of Gardner's model. Machine learning is the discipline of deriving actions or new insights from a (sufficiently large) collection of data. ML is used to derive an approximate solution that can be used to solve the problem for other unknown cases (Alpaydin, 2014, p. 1 ff.). The ML process roughly follows three steps. The first step is the *data input* step, in which data relevant to the problem is collected and processed. This data set is generally incomplete, as in practice the complete data space cannot be covered. This point is followed by step two, the *abstraction*. Here, the data is abstracted from its original form and passed to a model. The model is formed by appropriate procedures (e.g. training of the model) in such a way that it corresponds to the input data. In the last step the *generalization* takes place. The model should now be able to make decisions about an unknown data set (test data set). This is the reason why in general a predefined rule set is not sufficient. Instead, a heuristic approach is chosen, according to which the solution is approximated (Chandramouli & Dutt, 2018). There are multiple ways to implement ML, the most common being supervised, unsupervised and reinforcement learning. Common models used in ML include artificial neural networks or decision trees (Alpaydin, 2014).

### 5.1.1. ML VIA BRUTE FORCE

Now let us look at how HER generates its rules. The rules to move a piece in a given position are "learned" by testing **all** possible states of the game and eliminating moves if the result is undesirable. However, this brute force approach contradicts the general approach of ML, in that information is derived from a finite data set to lead to generalized behavior. This brute force method is also why Hexapawn does not scale well to more complex scenarios (4x4 or 8x8 squares)

when the amount of needed matchboxes becomes impracticable. HER thereby illustrates well, why we need ML for these kinds of problems, but represents it poorly itself. It can also contribute to the misconception H2 of (Heuer et al., 2021). Thus, one of the central problems of ML, data preparation, is downplayed by portraying the process as the collection of all possible data.

### 5.1.2. TRAINING WITH NEGATIVE EXAMPLES

The second misconception arises from the learning process and in particular the type of data HER receives. The learning process of removing beads characterizes both reinforced learning and ML deficient. In practice reinforced learning also relies on reinforcement of behavior. However, by removing beads in the original idea, only punishment is addressed (which is also less used in practice). While HER uses negative examples to remove unwanted behavior, in classical ML tasks behavior is trained using positive examples. Thus, while the learning procedure of HER is not categorically wrong, the student may get the wrong impression, which is that knowledge or behavior is built by removing undesirable behavior through negative examples rather than generalizing behaviour through example data in a ground-up process. A solution described by (Opel et al., 2019) and (Lindner & Seegerer, 2020) is already possible in Gardner's model by adding beads for winning moves.

## 5.2. Misconceptions of Models in ML

The second type of possible misconceptions arises from the context in which HER is used. As analyzed in the reception and adaptions section, HER is mostly used as a simple example of ML processes. Following on from this, specific models of ML are introduced. Since learners are mostly ML novices and the structure of HER is usually not analyzed in detail, there is a chance that students intuitively interpret Hexapawn as one of the specified models. However, due to the fact that Hexapawn does not adequately represent these, misconceptions about individual models of ML may be formed.

### 5.2.1. ML IN EXPERT SYSTEMS

As already analyzed, HER is a rule-based expert system. These, however, are generally not based on methods that conform to the classical idea of ML (which is supposed to be transported with HER), because they lack the possibility to learn from input Data and to adapt their fact base (Ogidan et al., 2018). There are rarely approaches in which either a hybrid method is used to train a ML model and then post-process the result by an expert system (Villena-Román et al., 2011), or the rule base is derived inductively from facts (Weiss & Indurkhya, 1995). Not only are these approaches exceptions, but also is neither of them presented

by HER. Therefore, HER is suitable for the representation of a rule-based expert system, but not in combination with ML. In addition, the misconception **(M3)** may arise, that ML on rule based expert systems is the systematic removal of false facts by negative examples which builds on the general misconception **(M2)**. (Lindner & Seegerer, 2020) use Hexapawn elsewhere in their material as an example of an expert system. Here the use is justified and a good example with suitable didactic reduction.

### 5.2.2. ML CHESS ENGINES AND DECISION TREES

The use of a "brute-force" method and the presentation of the game provide further opportunities for the formation of misconceptions. A classical chess engine generally consists of a heuristic function and a search tree optimization. While the search tree optimization is a classical optimization task, a ML approach can be used for the heuristics. The trained model receives the current position as input and outputs an evaluation of the position. Thus, the mode of operation is fundamentally different from HER and cannot be derived from it **(M1)(M2)**. This would at the same time reinforce the misconception **(H4)** formulated by *Heuer et al*, since the complexity of chess engine engineering is disregarded. It is true that the learning outcome of HER is critically dependent on the strength of the player. However, this does not sufficiently simulate the extent to which AI engineers are necessary for ML success, in that they are especially involved in modeling and data preparation rather than in the active learning process **(H2)**. (The Royal Institution, 2008) try to make the leap from HER to chess by addressing the game tree that results over several moves in a game of Hexapawn, referring to it as a decision tree. This is problematic in several ways. A decision tree is a common model in ML but is not to be confused with the game tree of a chess game. Furthermore, HER does not adequately represent either a game tree (the white moves are missing) or a decision tree nor does HER use a decision tree to evaluate a position. In fact, each position is considered independently of its successors. Therefore, HER is not suited to represent a decision tree, which in turn enables the formation of a misconception **(M4)** regarding decision trees. (Lindner & Seegerer, 2020) take a different approach by replacing the pawns with crocodiles and monkeys. This weakens the similarity to chess and tries to avoid listed misconceptions.

### 5.2.3. ML AND NEURAL NETWORKS

Lastly, it should be noted that in the investigated material artificial neural networks (ANN) are also often mentioned. At this point it is important to point out that HER is not a good model for these networks either, since none of its components are represented by the model. Nevertheless, especially inexperienced students (for whom the material is designed) can make a connection from HER to ANN, since, on the one hand, ML is used almost synonymously with ANN in public media and. On the other hand, due to the structure of interconnected items (matchboxes) and the weighting of the game paths (beads), there is a possibility that students will make a connection to the neurons and weights of ANN. HER, thus, should explicitly be differentiated from ANN.

## 6. Conclusion

HER is very motivating especially for young students due to its simple rules, illustrative learning process and playful character. Therefore, it is a suitable tool to convey the idea of ML (in that a computer can learn and adapt its behavior through new information) or teaching the concept of reinforced learning. Nonetheless, based on the previous analysis students may develop misconceptions about the general working of ML or important models used in ML with Hexapawn. Further investigations are necessary to determine whether and to what extent the described misconceptions are developed when playing Hexapawn and which adaptions have to be made to the material to circumvent these.

Additionally with the aforementioned considerations in mind, the question must be raised whether individual aspects of ML can be better conveyed through other unplugged materials avoiding listed misconceptions, as there are already materials which clearly convey the basic concepts of ML without sacrificing formal accuracy. With the material "Train a Neuron" by *cs4fn*[4], for example, a similar haptic feeling can be created with a game board and coins, but with the advantage that both the structure, functioning and learning procedures of a Neuron are taught. With the material of *Blum & Girschick* it is possible to teach (un)supervised learning with pen and paper only[5]. If one wants to focus on reinforcement learning, *Blum* developed a learning activity for this as well[6]. If, on the other hand, one does not want to forego the approach via a game, "Brain in a Bag", also by *cs4fn*[7], is a another choice. With this material, the game Snap or, with slight adaptations, games such as Nods and Crosses can be addressed. This way it is also possible to simulate a classic game AI. Finally, it should be pointed out that HER is also suitable for teaching a classical AI method: the rule-based expert system (as described by (Lindner & Seegerer, 2020)).

---

[4] https://cs4fndownloads.wordpress.com/train-a-neuron/

[5] https://explore.iteratec.com/blog/machine-learning-tutorial-teil-1 (german)

[6] https://explore.iteratec.com/blog/machine-learning-tutorial-teil-2 (german)

[7] http://www.cs4fn.org/teachers/activities/braininabag/braininabag.pdf

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
