# OpenReview forum: "Using Matchboxes to Teach the Basics of Machine Learning: an Analysis of (Possible) Misconceptions"
_ecmlpkdd.org/ECMLPKDD/2021/Workshop/TeachML — TeachML 2021_

### Official Review · Reviewer_1atP · 2021-07-16
**Review paper on uses of matchboxes for Hexapawn games in teaching ML content**

**Rating:** 6
**Confidence:** 3

**Review:**

The paper discussed an "unplugged" example of data-based learning, namely rule-based algorithm to play Hexapawn and discusses its strengths and weaknesses for teaching ML.

The paper starts with a valuable literature-based discussion of misconceptions in pedagogy providing a clear framework for discussing misconceptions as a trait in learning processes. Further, Hexapawn and the Matchbox-based trraining process is explained before the use and reception of the game's use in ML-teaching is discussed and then critiqued.

The discussion is interesting, but would benefit from addressing a few comments:

* H1-H4 would beneffit from contrasting the listed misconception to their "scientifically correct"  statements. A number of rebuttals come to mind that would be interesting to address
H3: a number of examples exist where neural networks can pick up on imperceptible patterns to make their predictons (adversaial examples emerged as a field for this idea)
H4: products such as AutoML do attempt to lower the bar for using ML techniques considerably
H2: self-play is an important topic in reinforcement learning
H1: "Lifelong learning" attemps to integrate  new data conntinously

It would be useful to more clearly delineate misconceptions M2 and M3. It should be made clear that M1-M4 are ew hypothesised misconceptions that the  authors claim are common. How were M1-M4 found? Is this based on empirical data?

Exhaustive/brute force learning is discussed a few times, but it is not explained when HER is fully trained, Sec 3. gives a range of number of games needed, therefore presumably learning occured before all possible states are encountered

The claim that it's easy to mistake matchboxes and beads as neurons and weights is surprisig. Is this reflected  in data?

---

### Official Review · Reviewer_MY2C · 2021-07-16
**Good survey/meta-analysis paper about misconceptions on teaching ML using pen-and-paper methods**

**Rating:** 8
**Confidence:** 4

**Review:**

This paper is a kind of review or meta-analysis of the Hexapod Educable Robot (HER) and possible misconceptions that are learned by students while learning using this tool.

The paper is very well written, and ideas flow clearly, and there is a good use of figures to explain the basic game using Matchboxes. I think there are only very minor writing issues such as reinforced learning -> reinforcement learning.

The paper discusses misconceptions that students can learn while using the HER and other similar teaching methods, it goes in very detailed description of misconceptions, both using its own categorization (The M's in Page 3) and Heuer's categorization at the end of the introduction.

I believe that this paper could be a good introduction to "pen and paper" ML teaching techniques, as it has plenty of references and use cases for these methods, and even extends into neural networks in Section 6, which is one of the limitations of HER. This paper is definitely valuable for this workshop, as I believe that most people are not aware of these "pen and paper" teaching methodologies and could be quite useful for teaching during the current pandemic.

---

### Decision · Program_Chairs · 2021-07-21

**Decision:**

Accept

**Comment:**

Congratulations! The reviewers agree that this paper should be accepted.

Camera-ready version is due August 18, 2021. As you prepare the camera ready version, please take the reviewers comments into consideration.

We look forward to your participation at the workshop on September 13, 2021. We invite you also to join us for the satellite event on September 08, 2021. Schedules for both the workshop and the satellite event will be forthcoming.